# Dissecting the Binding Affinity of Anti-SARS-CoV-2 Compounds to Human Transmembrane Protease, Serine 2: A Computational Study

**DOI:** 10.3390/ijms26020587

**Published:** 2025-01-11

**Authors:** Yue-Hui Shi, Jian-Xin Shen, Yan Tao, Yuan-Ling Xia, Zhi-Bi Zhang, Yun-Xin Fu, Ke-Qin Zhang, Shu-Qun Liu

**Affiliations:** 1State Key Laboratory for Conservation and Utilization of Bio-Resources in Yunnan and School of Life Sciences, Yunnan University, Kunming 650091, China; 13970055270@163.com (Y.-H.S.); jianxinshen0806@gmail.com (J.-X.S.); taoyan@ynu.edu.cn (Y.T.); xiayl@ynu.edu.cn (Y.-L.X.); kqzhang1@ynu.edu.cn (K.-Q.Z.); 2Yunnan Key Laboratory of Stem Cell and Regenerative Medicine, Biomedical Engineering Research Center, Kunming Medical University, Kunming 650500, China; zhangzhibi@kmmu.edu.cn; 3Human Genetics Center and Department of Biostatistics and Data Science, School of Public Health, The University of Texas Health Science Center, Houston, TX 77030, USA; yunxin.fu@uth.tmc.edu

**Keywords:** SARS-CoV-2, TMPRSS2, binding affinity, electrostatic effects, protein–ligand interactions

## Abstract

The human transmembrane protease, serine 2 (TMPRSS2), essential for SARS-CoV-2 entry, is a key antiviral target. Here, we computationally profiled the TMPRSS2-binding affinities of 15 antiviral compounds. Molecular dynamics (MD) simulations for the docked complexes revealed that three compounds exited the substrate-binding cavity (SBC), suggesting noncompetitive inhibition. Of the remaining compounds, five charged ones exhibited reduced binding stability due to competing electrostatic interactions and increased solvent exposure, while seven neutral compounds showed stronger binding affinity driven by van der Waals (vdW) interactions compensating for unfavorable electrostatic effects (including electrostatic interactions and desolvation penalties). Positive and negative hotspot residues were identified as uncharged and charged, respectively, both lining the SBC. Despite forming diverse interactions with compounds, the burial of positive hotspots led to strong vdW interactions that overcompensated for unfavorable electrostatic effects, whereas negative hotspots incurred high desolvation penalties, negating any favorable contributions. Charged residues at the SBC’s outer rim can reduce binding affinity significantly when forming hydrogen bonds or salt bridges. These findings underscore the importance of enhancing vdW interactions with uncharged residues and minimizing the unfavorable electrostatic effects of charged residues, providing valuable insights for designing effective TMPRSS2 inhibitors.

## 1. Introduction

Since its emergence in Wuhan, China, in late 2019, severe acute respiratory syndrome coronavirus 2 (SARS-CoV-2), the causative agent of coronavirus disease 2019 (COVID-19), has rapidly spread worldwide, profoundly impacting public health, economics, and societies [1,2]. As of July 2024, the WHO [3] had reported over 775 million confirmed cases and nearly 7 million deaths globally. Following earlier outbreaks, such as SARS-CoV in 2002 [4] and MERS-CoV in 2012 [5], this pandemic underscores the ongoing need for research into SARS-CoV-2 pathogenesis and the development of effective therapeutics and vaccines, even as the WHO declared an end to the public health emergency of international concern related to SARS-CoV-2.

SARS-CoV-2 is a positive-sense single-stranded RNA virus with an approximately 30 kb genome [6]. Like other coronaviruses, it utilizes the spike (S) glycoprotein for host cell entry [7,8]. The S protein consists of two subunits: S1 and S2 [9,10]. The S1 subunit’s receptor binding domain (RBD) engages with angiotensin-converting enzyme 2 (ACE2) on the cell membrane [11,12]. The S2 subunit mediates viral–cell membrane fusion through conformational changes that occur following S1 shedding [13,14].

The membrane fusion entry pathway involves S protein activation through cleavage at two sites, S1/S2 and S2’, by the host proteases Furin [15] and TMPRSS2 [16], respectively. Furin cleavage at S1/S2 sheds the S1 subunit, exposing the S2 subunit [11]. TMPRSS2 then cleaves at S2’, releasing the fusion peptide, which integrates with the target cell membrane to initiate membrane fusion [11,13]. Interrupting these cleavage events can effectively block viral entry. For a comprehensive review of the viral entry mechanisms and associated proteins, refer to [13,17,18].

Vaccine-elicited antibodies prevent infection by blocking S1-ACE2 attachment [19,20] or interfering with S2-mediated fusion [21]. However, mutations can give rise to antibody-escaping variants [22]. Targeting host proteases like Furin and TMPRSS2 offers an alternative approach [23,24], with several advantages. First, human proteases mutate less frequently than the viral S protein, reducing resistance risk. Second, protease inhibitors can combat multiple virus variants that share the same entry mechanism. Third, these inhibitors offer both prophylactic and therapeutic potential [25], unlike prevention-focused vaccines.

TMPRSS2 presents a more promising antiviral target than Furin for several reasons. First, TMPRSS2 is concentrated on the surface of respiratory and gastrointestinal epithelial cells [26], the main sites of infection, whereas Furin is widely distributed across tissues, reducing target specificity [27]. Second, inhibiting TMPRSS2 may result in fewer side effects due to its potential functional redundancy [28,29], while Furin’s essential role across many physiological processes [27] suggests a higher risk of adverse effects from its inhibition. Finally, TMPRSS2 inhibition has been shown to substantially impede viral entry, especially in highly transmissible variants like Omicron [30,31], whereas Furin inhibition only partially reduces SARS-CoV-2 infectivity [32,33].

TMPRSS2, a type II transmembrane serine protease, comprises three regions: an N-terminal intracellular domain, a single-pass transmembrane region, and an extracellular domain [34]. The extracellular domain includes three subdomains: a low-density lipoprotein receptor type-A (LDLR-A) domain, a scavenger receptor cysteine-rich (SRCR) domain, and a C-terminal trypsin-like serine protease (SP) domain [35]. The SP domain contains the canonical Ser441–His296–Asp345 catalytic triad, which is responsible for TMPRSS2’s enzymatic activity [36,37]. Currently, the experimentally resolved TMPRSS2 structures in the Protein Data Bank (PDB) represent only the extracellular domain, likely due to the technical challenges of resolving membrane protein structures, particularly the transmembrane region. Additionally, the intracellular domain has been predicted by AlphaFold2 [38] to be intrinsically disordered with very low confidence scores, further complicating structural characterization. Appendix A illustrates a representative experimental structure of the extracellular domain alongside the full-length AlphaFold2-predicted structure, highlighting these differences.

Existing drugs such as camostat mesylate, nafamostat mesylate, and bromhexine hydrochloride have shown efficacy in blocking SARS-CoV-2 infection in vitro and in animal models [39,40,41]. With their established safety profiles and favorable pharmacokinetics, these drugs are attractive candidates for repurposing as COVID-19 treatments, though efficacy in humans awaits confirmation in clinical trials.

Beyond these repurposing candidates, additional TMPRSS2 inhibitors have been identified from diverse sources. Peiffer et al., using in silico and biochemical methods, discovered three noncovalent inhibitors (debrisoquine, pentamidine, and propamidine) that effectively block SARS-CoV-2 infection in cellular models [42]. Chen et al. screened a library of 2560 FDA-approved and clinical trial compounds, identifying six compounds (cilnidipine, dasatinib, halofuginone, homoharringtonine, venetoclax, and verteporfin) with potent anti-SARS-CoV-2 activity via a TMPRSS2-dependent mechanism [43]. Huang et al. identified six TMPRSS2-targeting compounds (apigenin, baicalein, luteolin, naringenin, scutellarin, and wogonin) from *Scutellaria barbata* D. Don extracts, all of which have shown efficacy in inhibiting viral infection [44].

Despite the growing number of identified TMPRSS2 inhibitors, we have focused on these 15 compounds with strong experimental evidence for in-depth mechanistic exploration. We acknowledge the limitations of concentrating on a subset in this rapidly evolving field.

Although previous studies have explored the interactions and inhibition mechanisms of specific TMPRSS2 inhibitors [45,46,47,48,49], the complexities of binding and determinants of affinity remain only partially understood. This study employs a comprehensive suite of computational methods, including molecular docking, MD simulation, free energy landscape (FEL) reconstruction, binding free energy (BFE) calculation, and binding mode analysis, to dissect the binding affinities of the 15 TMPRSS2 antiviral compounds. Our findings reveal the following: (1) the mechanistic basis for binding affinity differences among compounds and (2) the fundamental principles governing high-affinity binding, offering essential insights for the rational design of future TMPRSS2 inhibitors.

## 2. Results

### 2.1. Molecular Docking Scores

Fifteen compounds were docked into the TMPRSS2 active site (i.e., the SBC) to generate protein–ligand complex structures. Table 1 lists each compound’s identifier, name, ZINC ID, and optimal docking score, with two-dimensional (2D) representations shown in Appendix A.

As shown in Table 1, docking scores range from −6.94 (C12; luteolin) to −4.17 (C7^+^; homoharringtonine) kcal/mol, where more negative scores indicate stronger predicted binding affinities. However, these scores should be considered as preliminary indicators, as scoring functions often oversimplify the complexity of protein–ligand interactions [50,51]. Further analyses, including MD simulations and BFE calculations, are necessary to validate these docking results.

### 2.2. Binding Stability

The 15 docked complexes underwent 100 ns MD simulations to assess structural and binding stability. Figure 1 shows the root mean square deviation (RMSD) profiles for the TMPRSS2 C_α_ atoms and ligand-heavy atoms relative to their starting structures. Twelve complexes maintained RMSD values below 0.5 nm throughout the simulation (Figure 1A), indicating small deviations. These complexes reached equilibrium within 2 ns and exhibited stable RMSD fluctuations, suggesting robust ligand binding.

Conversely, three complexes, TMPRSS2-C4, TMPRSS2-C7^+^ (Figure 1B), and TMPRSS2-C1 (Figure 1C), exceeded RMSD values of 0.5 nm. TMPRSS2-C4 reached equilibrium after approximately 5 ns, fluctuating between 0.3 and 0.6 nm. TMPRSS2-C7^+^ showed drastic fluctuations during equilibrium (13–88 ns), then jumped to 0.8 nm around 90 ns, persisting until the end of the simulation. TMPRSS2-C1^+^ showed a continuous RMSD increase, surpassing 4.0 nm after 55 ns.

RMSD values exceeding 0.5 nm signify substantial conformational changes from the equilibrated docked complex structure, suggesting significant ligand displacement. Large RMSD fluctuations further indicate complex instability, possibly due to ligand relocation or dissociation.

To verify the reproducibility of structural and binding stability, we conducted two additional 100 ns replicate MD simulations for each complex, both initiated from the same equilibrated conformation but with distinct initial atomic velocities assigned from a Maxwell distribution at 310 K. For each complex, the resulting RMSD profiles (Appendix A) exhibit consistent trends in values and fluctuations with those from the first simulation, confirming the reliability and robustness of our observations. Given these consistent results, we designated the first simulation as the representative replicate for subsequent structural superimposition to streamline the analysis.

Figure 2 presents superimposed structural frames at five time points from the MD trajectory for each complex. Compounds C10, C11, C12, C13, and C15 demonstrate stable binding within TMPRSS2’s SBC, with relatively small shifts in both location and conformation. In contrast, C2, C3, C5^+^, C6^+^, C8^+^, C9^−^, and C14^−^ exhibit larger variations but stay within the SBC.

C1^+^ begins to exit the SBC at 2 ns and fully detaches by 4 ns, while C4 exits at 30 ns but remains tethered by binding to an adjacent groove. C7^+^ leaves the SBC after 90 ns but stays associated with TMPRSS2. These observations corroborate the RMSD-based analyses, visually confirming compound stability and displacement patterns.

Since C1^+^, C4, and C7^+^ exited the SBC during simulations, they are unlikely to compete effectively with the substrate and were excluded from further analyses.

### 2.3. FEL

For the remaining 12 complexes, FELs were reconstructed from the equilibrated portions (5–100 ns) of the MD trajectories. Figure 3 illustrates the 2D FELs, with reaction coordinates defined by the buried surface area (BSA) between TMPRSS2 and the compound, and the interface RMSD (iRMSD) representing the binding interface space. As shown in Figure 3, TMPRSS2-C9^−^ covers the largest area in this space, indicating the largest interface entropy. Medium areas are observed for TMPRSS2-C2, -C3, -C5^+^, -C6^+^, and -C8^+^, while the smallest areas are seen for TMPRSS2-C10 to -C15.

TMPRSS2-C11, -C13, and -C15 exhibit funnel-like FELs with a single global minimum and almost no local minima, suggesting stable binding interfaces with limited conformational diversity. Complexes such as TMPRSS2-C3, -C6^+^, -C8^+^, and -C10 contain one free energy basin, while TMPRSS2-C2, -C5^+^, and -C9^−^ feature two basins. These basins, with energy levels below −8 kJ/mol, contain multiple local or global minima (typically 2–3), indicating several metastable or stable interface states sampled during MD simulations.

The global minima of TMPRSS2-C11, -C12, -C13, and -C15 (−18 kJ/mol) are lower than those of other complexes (−17 to −15 kJ/mol), indicating higher binding interface thermostability and stronger protein–ligand association. This observation partially aligns with the snapshot superimposition analysis.

Although the 12 complexes exhibit varied interface thermodynamics, the constructed FELs effectively identify the most energetically favorable interface conformations. To further assess BFE, 500 snapshots from each complex’s global free energy minimum were randomly selected to represent the structural ensemble for subsequent BFE calculations.

### 2.4. BFE

Table 2 presents the BFE (Δ*G*_binding_) and its energy components for the 12 complexes, calculated using the molecular mechanics Poisson–Boltzmann surface area (MM-PBSA) method. The average Δ*G*_binding_ values range from −87.6 (TMPRSS2-C2) to −12.9 kJ/mol (TMPRSS2-C8^+^), indicating C2 as the strongest binder and C8^+^ as the weakest. Notably, TMPRSS2-C6^+^, -C8^+^, -C9^−^, and -C14^−^ exhibit substantial standard deviations (SDs) in Δ*G*_binding_, exceeding or approaching the magnitude of their average values. This indicates high BFE variability, with multiple snapshots in the ensemble exhibiting positive values, signifying unfavorable binding.

For all complexes, both vdW interaction energy (Δ*E*_vdw_) and nonpolar solvation free energy (Δ*G*_nonpolar_, representing favorable solvent entropy gain from water displacement at the binding interfaces) contribute to lowering the BFE, with Δ*E*_vdw_ being the primary contributor to binding affinity. Conversely, the polar solvation free energy (Δ*G*_polar_), representing the energetic penalty of desolvating charged/polar groups upon binding, significantly reduces binding affinity due to its substantial positive values. All complexes, except TMPRSS2-C6^+^ and -C8^+^, have negative average Δ*E*_elec_ values, indicating a positive contribution of intermolecular electrostatic interactions to binding affinity.

Notably, TMPRSS2-C5^+^ has a Δ*E*_elec_ SD of 10 kJ/mol, which greatly exceeds its average value (−0.7 kJ/mol), indicating that electrostatic interactions frequently hinder binding affinity across many ensemble structures due to repulsion from the positive charges on both C5+ and TMPRSS2 (+1 and +5, respectively). Similarly, C6^+^ and C8^+^ also experience repulsion interactions with TMPRSS2, as reflected in their positive Δ*E*_elec_ values.

In contrast, the strongest electrostatic attractions are observed in complexes with negatively charged C14^−^ and C9^−^. However, these two complexes also experience the largest electrostatic desolvation penalties (Δ*G*_polar_), significantly counteracting their favorable Δ*E*_elec_ values. Consequently, the overall net electrostatic effect, defined as the sum of Δ*E*_elec_ and Δ*G*_polar_, is more detrimental to binding affinity in these complexes compared to those with neutral compounds.

For TMPRSS2-C6^+^ and -C8^+^, favorable vdW interactions are insufficient to counteract unfavorable electrostatic effects, highlighting the critical role of solvent entropy gain (Δ*G*_nonpolar_) in lowering BFEs. For TMPRSS2-C5^+^, -C9^−^, and -C14^−^, while favorable vdW interactions can compensate for the unfavorable electrostatic effects, Δ*G*_nonpolar_ remains essential in lowering their BFEs.

Overall, all five charged compounds consistently exhibit weaker affinities for TMPRSS2 compared to neutral ones (Table 2). This is primarily because their vdW interactions cannot fully counterbalance the net unfavorable electrostatic effects, regardless of differences in solvent entropy gain.

### 2.5. Residue Binding Free Energy (RBFE) Decomposition

Figure 4 illustrates the per-RBFE values for TMPRSS2 in complex with the 12 compounds. TMPRSS2 bound to charged compounds shows more residues with significant RBFE values (exceeding ±3.0 kJ/mol) compared to when it is bound to neutral compounds. These influential residues, primarily charged (Appendix A), are distributed throughout TMPRSS2, highlighting the critical role of long-range electrostatic interactions in modulating RBFE.

For TMPRSS2 bound to positively charged compounds (C5^+^, C6^+^, and C8^+^), positively charged residues (Arg and Lys) reduce binding affinity (positive RBFE values), while negatively charged residues (Asp and Glu) enhance it (negative RBFE values). Conversely, for TMPRSS2 bound to negatively charged compounds (C9^−^ and C14^−^), these effects are reversed.

Nevertheless, the overall BFE is determined by the cumulative effect of all individual RBFEs, rather than the contribution of any single residue. This is particularly evident in complexes with charged compounds, where opposing negative and positive RBFE values largely offset each other, resulting in less favorable total BFEs compared to those with neutral compounds (Table 2).

For TMPRSS2 bound to neutral compounds, most residues exhibit RBFE values near 0 kJ/mol, with only a few exceeding ±1.0 kJ/mol. Residues with RBFEs exceeding ±3.0, between ±1.0 and ±3.0, and within ±1.0 kJ/mol are classified as having a significant, moderate, and weak/negligible impact on binding affinity, respectively. Notably, residues with significant and moderate effects on affinity are mainly located within or near the SBC, comprising five pockets/subsites: S4, S3, S2, S1, and S1’ (Appendix A) [37,45,52].

Asp435 exhibits the largest positive RBFE values in five of the seven neutral compound complexes (TMPRSS2-C2, -C3, -C10, -C12, and -C13; Figure 4 and Appendix A), indicating its prominent role in reducing affinity. Its affinity-reducing effects are moderate in TMPRSS2-C15 (2.4 kJ/mol; Appendix A) and weak in TMPRSS2-C11 (0.2 kJ/mol).

Glu389 exhibits the strongest affinity-reducing effects in complexes with C15 and C11 and weakly reduces affinity for most other neutral compounds, though it slightly enhances affinity for C3. Lys342 reduces affinity significantly or moderately for most neutral compounds, except for C15, where it has a weak enhancing effect. Asp345, Lys390, and Lys467 each significantly reduce affinity for one specific neutral compound, with effects on the others ranging from moderate to weak enhancement or reduction.

Gln438 exhibits the strongest affinity-enhancing effects for C3, C11, C12, and C15, with the most negative RBFE values (Appendix A). It also significantly enhances the affinities of other neutral compounds and all charged compounds except C9^−^, where it slightly reduces affinity. Trp461 has the strongest affinity-enhancing effects for C6^+^, C9^−^, C10, and C13, while Cys465 shows this effect for C2. Both residues generally enhance the affinities of other compounds significantly or moderately. Cys437, Gly462, and Ser463 typically exhibit significant or moderate enhancing effects across most compounds. Although Cys297, Leu302, and Gly464 often exert weak effects, they can significantly or moderately enhance affinity for certain compounds (Appendix A).

In summary, charged TMPRSS2 residues significantly influence the binding affinities of all charged compounds through long-range electrostatic interactions. Within or near the SBC, the charged residues typically weaken the affinities of neutral compounds. Asp435 and Lys342, in particular, substantially reduce binding affinity for most neutral compounds and are therefore identified as “negative hotspots”. Conversely, several uncharged residues, namely Gln438, Trp461, Cys465, Cys437, Gly462, and Ser463, strongly enhance binding affinity for both charged and neutral compounds, classifying them as “positive hotspots”.

### 2.6. Binding Modes and RBFE Components

To explore how binding modes influence binding affinity, we conducted a comprehensive analysis of interaction patterns and RBFE components for each TMPRSS2–compound complex.

#### 2.6.1. Intermolecular Interactions

Figure 5 shows the representative 2D binding modes of 12 complexes, each derived from a snapshot with a BFE value closest to the average. Across all these complexes, TMPRSS2 residues predominantly interact with compounds via vdW contacts (light green). Additionally, alkyl–alkyl and pi–alkyl interactions (light pink), which are classified as vdW interactions due to their basis in London dispersion forces [53,54], are frequently observed. These findings align with BFE calculations, collectively indicating that vdW forces are the primary contributors to enhancing the TMPRSS2-binding affinities of these compounds.

The 12 complexes exhibit a range of electrostatic-related interactions, including salt bridges, conventional hydrogen bonds (HBs), carbon HBs, pi-related stacked (amide–pi and pi–pi), pi–sulfur, and pi–anion interactions. Table 3 summarizes the number of occurrences of each interaction type.

Salt bridges are observed exclusively in complexes with negatively charged compounds: C9^−^ with Lys300 and C14^−^ with Lys390 (Figure 5 and Table 3). Conventional HBs range from none in TMPRSS2-C9^−^, -C11, and -C12 to nine in TMPRSS2-C2. Carbon HBs are present in most complexes except those with C3 and C15. Pi-related interactions are common, with Trp461 and Cys437 frequently involved (Figure 5). Uniquely, C9^−^ has pi–anion interactions between its porphyrin ring and Glu299.

Despite these diverse electrostatic-related interactions, their quantity and type (Table 3) do not correlate with the overall electrostatic strength (Δ*E*_elec_; Table 2). In complexes with charged compounds, strong electrostatic attraction or repulsion is primarily determined by long-range forces, which dominate over short-range interactions like salt bridges, HBs, or pi-related contacts. Thus, the overall electrostatic strength reflects a balance between these distant forces and localized interactions.

#### 2.6.2. Correlating Hotspot Interactions with RBFE

Gln438, a key positive hotspot enhancing the binding of most compounds (except C9^−^), interacts with ligands via vdW contacts (TMPRSS2-C3, -C5^+^, -C9^−^, -C10, C11, C13, C15), carbon HBs (TMPRSS2-C2, -C6^+^, -C8^+^, -C12), or conventional HBs (TMPRSS2-C14^−^). Situated at the S3 subsite within the SBC, Gln438’s side chain protrudes outward (Appendix A) but becomes buried upon ligand binding (Figure 5).

A comparison of Gln438’s energy components across complexes reveals that increased burial corresponds to higher desolvation penalties, lower vdW interaction energies, and greater solvent entropy gains, with minimal effect on electrostatic interaction energies (Appendix A). In most complexes, strong vdW interactions outweigh the net unfavorable electrostatic effects, resulting in significantly low RBFE values for Gln438. However, in TMPRSS2-C9^−^, Gln438’s limited burial reduces the desolvation penalty, but the weak vdW interactions fail to compensate, resulting in a positive RBFE value that weakens binding.

Trp461, another key hotspot located at the S4 pocket’s bottom (Appendix A), interacts with compounds via vdW contacts, carbon HB, pi–pi stacked, or alkyl–alkyl/pi–alkyl interactions (Figure 5). Increased burial of Trp461 primarily strengthens vdW interactions over electrostatic interactions, as observed in TMPRSS2-C5^+^, -C6^+^, -C8^+^, -C9^−^, -C10, and -C13 (Figure 5 and Appendix A). Although stronger electrostatic interactions typically incur higher desolvation penalties, the substantial increase in vdW strength indicates that vdW forces dominate Trp461’s affinity-enhancing effect.

Cysteine hotspots Cys465 and Cys437 interact with compounds via pi–alkyl, pi–sulfur, amide–pi stacked, or vdW interactions. Cys437, located on the S1 pocket’s side wall, exhibits stronger vdW and often stronger electrostatic interactions than Cys465, which is positioned at S1’s outer rim (Appendix A). However, due to its more buried location, Cys437 incurs greater desolvation penalties. Despite these penalties, vdW interactions dominate for both cysteines, outweighing net electrostatic effects, whether favorable or unfavorable, to enhance binding affinity.

Two additional hotspot residues, Gly462 and Ser463, are located at the outer rims of the S1/S4 and S4 pockets, respectively (Appendix A), and form conventional HBs and amide–pi stacked interactions with specific compounds (Figure 5). Although their electrostatic effects are often unfavorable, strong vdW interactions compensate for these effects.

Asp435 and Lys342, located at the bottom of the S1 pocket and the outer rim of the S2/S4 pocket, respectively, act as negative hotspots for neutral compounds. Asp435 forms conventional HBs with the amidine groups in C2 and C3 and the hydroxyl group in C10, generating strong electrostatic interactions but incurring substantial desolvation penalties. These penalties persist even when Asp435 does not directly contact other neutral compounds in the representative binding modes, continuing to reduce binding affinity. A similar effect is observed with Lys342.

#### 2.6.3. Compound Solvent Exposure

Charged compounds (C5^+^, C6^+^, C9^−^, and C14^−^) generally interact with fewer TMPRSS2 residues than neutral compounds, leaving more of their groups/atoms exposed to solvent (blue vignettes in Figure 5). This reduced interaction increases their mobility by minimizing structural constraints from the protein, thereby disrupting protein–ligand interactions and reducing binding stability. These effects are reflected in the increased conformational and locational shifts in charged compounds compared to neutral ones (Figure 2). Although C8^+^ interacts with more residues due to its larger size, many of its atoms remain solvent-exposed, which also weakens binding stability.

## 3. Discussion

Although all 15 compounds demonstrate TMPRSS2-dependent anti-SARS-CoV-2 activity [42,43,44], our MD simulations consistently reveal that compounds C1^+^, C4, and C7^+^ failed to stably occupy TMPRSS2’s SBC. These compounds also exhibit less negative docking scores compared to most others, suggesting weaker binding affinities within our docking framework. This observation raises the possibility that they may preferentially target allosteric sites or interact with TMPRSS2-associated regulatory proteins, rather than directly competing with the viral S protein at the active site.

The remaining 12 compounds stay within the SBC during simulations but exhibit varying degrees of conformational and locational shifts (Figure 2). Their complexes also show diverse thermodynamic properties at the binding interface (Figure 3). Generally, more pronounced shifts within the SBC correspond to higher interface entropy and richer interface diversity, both of which are associated with decreased binding stability.

However, some compounds deviate from this trend. For instance, C10 remains stable within the SBC, resulting in low interface entropy, but still exhibits two stable and one metastable interface state. Conversely, C14^−^ undergoes significant conformational and locational shifts, yet its complex maintains low interface entropy with 2–3 stable interface states. These seemingly contradictory observations can be attributed to the different metrics used: the structural superimposition analysis is based on the RMSD of global protein C_α_ atoms and ligand-heavy atoms, whereas FEL reconstruction focuses on local interface properties, using iRMSD and BSA as reaction coordinates.

Notably, in their complexes with TMPRSS2, the five charged compounds consistently exhibit large conformational and locational shifts, coupled with high binding interface entropy and diversity, indicating an unstable association with TMPRSS2’s active site. This instability is reflected in their higher BFEs and SDs compared to those of the neutral compounds complexed with TMPRSS2. Despite the BFE calculations being based on structural ensembles from the FEL’s global minima, which may under-represent the full spectrum of conformations, these findings nonetheless underscore the inherent instability of charged compounds.

Further RBFE decomposition reveals that charged residues in TMPRSS2 significantly impact the binding affinity of charged compounds (Figure 4). These residues exert long-range electrostatic forces on the compounds’ charged groups, disrupting their orientation and positioning through competing repulsive and attractive interactions. This disruption leads to an imbalance between electrostatic interaction energy and desolvation penalty. As shown in Table 2, charged compound complexes exhibit disproportionate changes in these energetic components compared to the more balanced relationship observed in neutral compound complexes. For example, TMPRSS2-C9^−^ and -C14^−^ exhibit stronger electrostatic interactions but higher desolvation penalties, while TMPRSS2-C5^+^ experiences weaker electrostatic attraction without a corresponding reduction in the desolvation penalty. Both TMPRSS2-C6^+^ and -C8^+^ experience electrostatic repulsion and high desolvation penalties. These imbalances amplify the unfavorable electrostatic effects, leading to lower binding affinities.

Energy component analysis reveals that vdW interactions play a dominant role in lowering both the total BFEs of complexes (Table 2) and the RBFEs of individual positive hotspot residues (Appendix A). The strength of vdW interactions effectively counterbalances the unfavorable electrostatic effects associated with desolvation penalties, thereby enhancing binding affinity. Additionally, although smaller in magnitude, nonpolar solvation free energy (i.e., solvent entropy gain) consistently contributes to enhancing binding affinity. This is particularly crucial in charged compound complexes, where strong unfavorable electrostatic effects effectively oppose the favorable vdW contributions.

Interestingly, solvent entropy gain correlates more strongly with vdW interaction strength than with electrostatic desolvation penalty, but not with the overall electrostatic interaction strength. This is primarily because increased vdW contacts lead to greater surface area burial, excluding more water molecules from binding interfaces and thereby enhancing solvent entropy gain. In contrast, electrostatic interactions, such as HBs and salt bridges, often remain partially solvent-exposed due to their dependence on polar or charged residues [55,56,57], contributing less to solvent entropy gain. Consequently, the nonpolar solvation free energy component can serve as an indicator of the extent of burial at the binding interface and, by extension, vdW interaction strength.

Nevertheless, variations in overall BFEs among complexes and in the RBFEs of different hotspots are primarily determined by the interplay between vdW interaction strength and the net electrostatic effect, which depends on the types and locations of specific electrostatic interactions at the binding interface.

Negatively charged compounds, such as C9^−^ and C14^−^, exhibit the strongest electrostatic interactions with TMPRSS2 due to long-range attraction (Table 2). However, the burial of polar surfaces from both the protein and ligand leads to substantial net unfavorable electrostatic effects, greatly counteracting favorable vdW contributions. A similar situation is observed with the negative hotspots Asp435 and Lys342, where the net unfavorable electrostatic effects arising from desolvation penalties outweigh vdW interactions (Appendix A), even in the presence of strong electrostatic interactions like conventional HBs. Asp435 has stronger affinity-reducing effects than Lys342 due to its greater desolvation penalties (Appendix A), attributed to its location deep within the S1 pocket (Appendix A). In contrast, Lys342, located at the outer rim of the shallower S2/S4 pocket, experiences less desolvation upon ligand binding.

Ser436, a non-hotspot residue located at the S1 pocket’s bottom alongside Asp435 (Appendix A), illustrates how side-chain polarity and energetic interplay dictate affinity-affecting effects through location. Despite forming conventional HBs with certain compounds (C2 and C3; Figure 5), Ser436’s less polar hydroxyl group incurs lower desolvation penalties than Asp435’s carboxylate, resulting in weaker net unfavorable electrostatic effects (Appendix A). Additionally, Ser436 experiences stronger vdW interactions (Appendix A) due to greater burial upon binding (Figure 5). Together, these factors explain why Ser436 has weaker affinity-affecting effects for neutral compounds.

Large, charged residues (Lys300, Glu389, and Lys390) and the small, uncharged Gly464 at the SBC’s outer rim have distinct effects on binding affinity, despite their involvement in strong electrostatic interactions. Specifically, Glu389 forms a conventional HB with C15’s hydroxyl group, while Lys300 and Lys390 form salt bridges with the carboxylate groups of C9^−^ and C14^−^, respectively. These interactions result in substantial burial of these residues (Figure 5), as indicated by the large magnitudes of nonpolar solvation free energy (Appendix A), leading to high desolvation penalties and, thus, strong net unfavorable electrostatic effects, which reduce binding affinity.

In contrast, Gly464, despite forming HBs with C2, C6^+^, and C14^−^, moderately enhances their affinities. This is due to Gly464’s small size and lack of a polar side chain, resulting in low desolvation penalties and, thus, weak net electrostatic effects, either favorable (as in TMPRSS2-C2) or unfavorable (as in TMPRSS2-C6+ and -C14^−^), with the unfavorable effects compensated for by vdW energies.

In all complexes, positive hotspot residues are consistently uncharged, enhancing binding affinity primarily through dominant vdW forces rather than electrostatic interactions (Appendix A). This dominance stems from their close contacts with ligand moieties and lower polarity compared to charged residues. Uncharged residues, whether polar or nonpolar, are generally less solvated than charged ones, resulting in smaller desolvation penalties regardless of the types of electrostatic-related interactions involved. The extent of burial, quantified by the magnitude of nonpolar solvation free energy (Appendix A) and depicted in Figure 5, directly correlates with vdW strength and solvent entropy gain. Consequently, increased burial of uncharged residues amplifies vdW interactions, outweighing the relatively minor increase in desolvation penalty. This relationship explains why positive hotspots are typically uncharged, while negative hotspots are charged, and why increased burial of uncharged residues boosts vdW interactions more effectively than it increases desolvation penalty.

Although intermolecular electrostatic interactions are generally considered crucial for binding affinity and stability [56,58,59,60], our study reveals a more nuanced picture. Surprisingly, residues involved in strong electrostatic interactions, such as salt bridges, can reduce binding affinity. Conversely, residues engaged in weaker electrostatic-related interactions, such as carbon HBs, pi-related stacked, and pi–sulfur interactions, tend to enhance affinity. Interestingly, residues forming conventional HBs do not necessarily enhance affinity. These findings emphasize the importance of considering the specific context of electrostatic interactions, including the nature of the interacting (partial) charges, their microenvironment, and structural location, when evaluating their role in ligand binding and drug development. While individual electrostatic interactions are stronger than individual vdW contacts and thus contribute significantly to binding stability, their complex, context-dependent impact on overall binding affinity underscores the need for a nuanced understanding of these interactions.

It is important to note that the TMPRSS2 construct used in this study excludes both the transmembrane region and the intracellular domain, potentially influencing the protein’s dynamics and ligand interactions. Nevertheless, prior studies [37,61] support the soluble extracellular domain as a reliable model for evaluating inhibitor binding. The absence of experimental binding affinity data for the 15 compounds currently limits direct benchmarking of our computational results, and future structural studies or biochemical assays will be essential to validate and extend these findings.

Despite these limitations, our study provides critical insights into TMPRSS2–inhibitor interactions, emphasizing a core principle: vdW interactions and electrostatic desolvation penalties are primary determinants of binding affinity. This principle can inform the design and optimization of inhibitors across diverse chemical scaffolds. Specifically, enhancing vdW interactions while minimizing unfavorable electrostatic effects provides a universal strategy for inhibitor development. These insights are highly relevant for advancing structurally varied inhibitors, including non-peptidic and fragment-based compounds.

## 4. Materials and Methods

### 4.1. Preparation of the TMPRSS2 Structure

The crystal structure of the TMPRSS2–nafamostat complex (1.95 Å resolution) was retrieved from the Protein Data Bank (PDB) [62] with PDB ID: 7MEQ [37]. Nafamostat, heteroatoms, and water molecules were removed. Missing atomic coordinates for residues 164–166, 203–207, 217–220, and 250–255 were added using the SWISS-MODEL sever [63], and the unresolved C-terminal residue Gly492 was modeled using PyMOL [64]. Hydrogen atoms were added using the H++ server [65], with protonation states for titratable residues assigned based on predicted pKa values at pH 7.4. AutoDockTools 1.5.6 [66] was used to merge nonpolar hydrogen atoms, assign Gasterier partial charges [67], and convert the file from PDB to PDBQT format.

### 4.2. Preparation of the Compound Structures

The structures of the 15 compounds in SDF format were retrieved from the ZINC15 database [68,69]. Open Babel 3.0.1 [70] was used to convert the files from SDF to PDB format. Avogadro 1.95 [71] was then employed to add hydrogen atoms at pH 7.4. Finally, AutoDockTools 1.5.6 was used to assign Gasteiger partial charges and convert the files to PDBQT format.

### 4.3. Molecular Docking

The 15 compounds were docked into TMPRSS2’s SBC using AutoDock 4.2 [66], based on the hypothesis that they inhibit enzyme activity by competing with the substrate for the active site. The docking grid was centered on the catalytic triad (Ser441–His296–Asp345) with dimensions of 60 Å × 60 Å × 60 Å and a grid spacing of 0.375 Å, ensuring the active site and surrounding regions were adequately encompassed.

During docking, the TMPRSS2 structure was kept rigid while full flexibility was allowed for ligand molecules. A conformational search was conducted using the Lamarckian genetic algorithm [72], with default mutation and crossover rates of 0.02 and 0.8, respectively. To ensure comprehensive sampling, 100 independent docking runs were conducted, each starting with a population of 150 individuals and allowing a maximum of 2,500,000 energy evaluations and 27,000 generations.

Docking poses were clustered with a RMSD threshold of 2.0 Å. The most favorable pose, selected from the largest cluster based on the lowest binding energy score, was converted from PDBQT to PDB format using Open Babel for subsequent MD simulation.

### 4.4. MD Simulation

MD simulations for protein–compound complexes were conducted using GROMACS 5.1.4 [73] to assess structural stability. The Amber ff99SB force field [74] was applied. The ligand parameters were generated using the Antechamber tool within AmberTools23 [75], with atomic partial charges calculated via the AM1-BCC method [76].

Each complex was solvated in a periodic dodecahedron box filled with TIP3P water [77], maintaining a minimum solute-to-wall distance of 12 Å. Na^+^ and Cl^−^ ions were added to neutralize the system and achieve a physiological salt concentration of 150 mM. Energy minimization was then performed using steepest descent and conjugate gradient methods until a maximum force of 10 kJ/mol/nm was reached.

System equilibration involved four consecutive 300 ps NVT simulations at 310 K with a 1 fs time step, progressively reducing harmonic restraints on the solute-heavy atoms (1000, 100, 10, and 0 kJ/mol/nm^2^). This was followed by a 300 ps NPT simulation at 310 K and 1 bar, without position restraints.

The production MD simulations were conducted with a 2 fs time step, using the LINCS algorithm [78] to constrain bond lengths. The Verlet cutoff scheme was used to manage neighbor lists with a 0.005 kJ/mol/ps buffer tolerance [79]. Long-range electrostatics were treated using the PME algorithm [80] with a 1.2 nm real-space cutoff. The Lennard-Jones potential for vdW interactions was truncated at 1.2 nm. Temperature and pressure were maintained at 310 K and 1 bar using the v-rescale [81] and Parrinello–Rahman algorithms [82], with coupling time constants of 0.1 and 2.0 ps, respectively. Each production run was performed for 100 ns, as our prior simulations [55,83] and related studies [84,85,86] have demonstrated that timescales up to 100 ns often strike a balance between computational cost and capturing key dynamic behaviors in the docked protein–ligand systems. Coordinates were recorded every 2 ps.

### 4.5. MD Trajectory Analysis

RMSD values for TMPRSS2 C_α_ atoms and ligand-heavy atoms were calculated relative to the starting structure using the GROMACS tool ‘gmx rms’ to assess structural stability and trajectory equilibrium. Subsequent analyses were restricted to the equilibrated portion of each trajectory to minimize artifacts. Protein–ligand minimum interatomic distance and solvent-accessible surface area (SASA) were computed using ‘gmx mindist’ and ‘gmx sasa’, respectively. Binding interface residues were defined as those within an average minimum distance of 4 Å from any ligand atom. The BSA upon binding was determined by subtracting the SASA of the complex from the combined SASA of the unbound ligand and protein.

### 4.6. FEL Reconstruction

The 2D FEL of each complex was reconstructed using the interface RMSD (iRMSD) and BSA as reaction coordinates. The iRMSD was calculated for the heavy atoms of the protein interface residues and the ligand. Free energy was determined using the following equation:(1)$$F\left(i\right)=-kTIn\left(\frac{P_{i}}{P_{max}}\right)$$
where $F\left(i\right)$ represents the free energy of state i, k is the Boltzmann constant, T is the temperature, $P_{i}$ is the probability of the system being in state i, and $P_{max}$ is the probability of the most probable state. FELs were generated using an in-house Python script (see Appendix A).

For each complex, 500 snapshots/structures were randomly selected from the FEL’s global free energy minimum as the representative ensemble for BFE calculations.

### 4.7. BFE Calculation

The MM-PBSA method [87] was employed to calculate the BFEs of the TMPRSS2–compound complexes. The BFE (${∆G}_{binding}$) was calculated as(2)$${∆G}_{binding}={∆E}_{MM}+{∆G}_{solvation}-T∆S=\left({∆E}_{bonded}+{∆E}_{vdW}+{∆E}_{elec}\right)+\left({∆G}_{polar}+{∆G}_{nonpolar}\right)-T∆S$$
where ∆*E*_MM_, ∆*G*_solvation_, and *T*∆*S* represent changes in vacuum MM potential energy, solvation free energy, and solute entropy upon protein–ligand binding, respectively. ∆*E*_MM_ includes bonded (Δ*E*_bonded_), vdW (Δ*E*_vdW_), and electrostatic (Δ*E*_elec_) energy changes. ∆*G*_solvation_ is divided into polar (Δ*G*_polar_) and nonpolar (Δ*G*_nonpolar_) components. Given the challenge of accurately estimating *T*Δ*S*, this term was excluded, as its absence has a negligible impact on comparing relative binding affinities across ligands with the same receptor [83,88].

MM-PBSA calculations were conducted using the GROMACS tool ‘g_mmpbsa’[89], with default parameters except for specific adjustments for Δ*G*_polar_ estimation: a grid resolution of 0.5 Å, an ionic strength of 0.15 M, a temperature of 310 K, and dielectric constants for the solute and solvent of 4 and 80, respectively.

Per- RBFE contributions were computed using the Python script ‘MmPbSaDecomp.py’ within g_mmpbsa, which also decomposed the RBEF into its energy components.

### 4.8. Binding Mode Analysis

For each complex, the snapshot with a BFE value closest to the ensemble average was selected as the representative structure. Protein–ligand binding modes in these representative structures were then analyzed using Biovia Discovery Studio [90].

## 5. Conclusions

Using computational methods, we investigated the binding affinity determinants of 15 compounds hypothesized to compete with the SARS-CoV-2 S protein for TMPRSS2’s active site. Three compounds exited the SBC during MD simulations, suggesting alternative inhibitory mechanisms. Of the remaining 12 compounds, the 5 charged ones exhibited reduced binding stability compared to the 7 neutral compounds, likely caused by competition between long-range electrostatic repulsion and attraction with TMPRSS2’s charged residues, and increased solvent exposure.

Intermolecular vdW interactions enhanced the overall binding affinity, while net electrostatic effects reduced it. Neutral compounds showed higher affinity as vdW interactions effectively compensated for unfavorable electrostatic effects, while charged compounds exhibited lower affinity due to an imbalance between electrostatic forces and desolvation penalties, which amplified the unfavorable electrostatic effects.

Positive and negative hotspot residues, which are uncharged and charged, respectively, are all located within the SBC. Upon ligand binding, the burial of positive hotspots resulted in strong vdW interactions that overcompensated for net unfavorable electrostatic effects, while the burial of negative hotspots incurred high desolvation penalties that outweighed any favorable contributions. Charged residues at the cavity’s outer rim can significantly reduce binding affinity when forming HBs or salt bridges due to substantial desolvation penalties.

In conclusion, (1) vdW interactions and desolvation penalties are the primary determinants of binding affinity. (2) Variations in binding affinity are governed by the interplay between vdW strength and net electrostatic effects, with nonpolar solvation free energy playing a crucial role in charged compound complexes. (3) Nonpolar solvation free energy, reflecting the burial extent of binding interfaces and individual residues, correlates more strongly with vdW interactions than with desolvation penalties. (4) A residue’s desolvation penalty depends on its polarity and burial extent upon binding, regardless of the specific types of interactions formed. (5) Although strong electrostatic interactions (e.g., HBs and salt bridges) can stabilize binding interfaces, they can also reduce affinity due to high desolvation penalties. Finally, we propose enhancing vdW interactions with uncharged residue lining the SBC while minimizing the unfavorable electrostatic effects from charged residues at the cavity’s outer rim as an effective strategy for future inhibitor optimization.

## Figures and Tables

**Figure 1 ijms-26-00587-f001:**
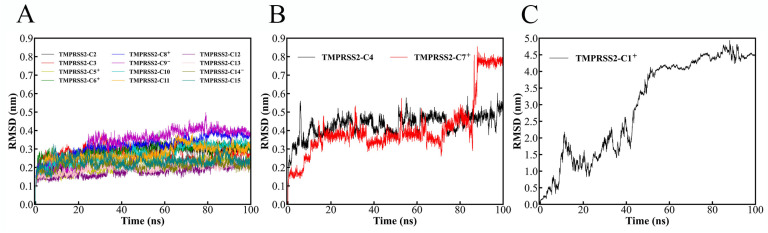
RMSD values of TMPRSS2 C_α_ atoms and compound heavy atoms in docked complexes during 100 ns MD simulations, relative to their equilibrated docked structures. (**A**) RMSD curves for complexes with values remaining below 0.5 nm throughout the simulation. (**B**) RMSD curves for TMPRSS2-C4 and -C7^+^, which exhibit large fluctuations and briefly exceed 0.5 nm after equilibrium. (**C**) TMPRSS2- C1^+^, which exhibits a continuous increase, consistently exceeding 0.5 nm after approximately 8 ns.

**Figure 2 ijms-26-00587-f002:**
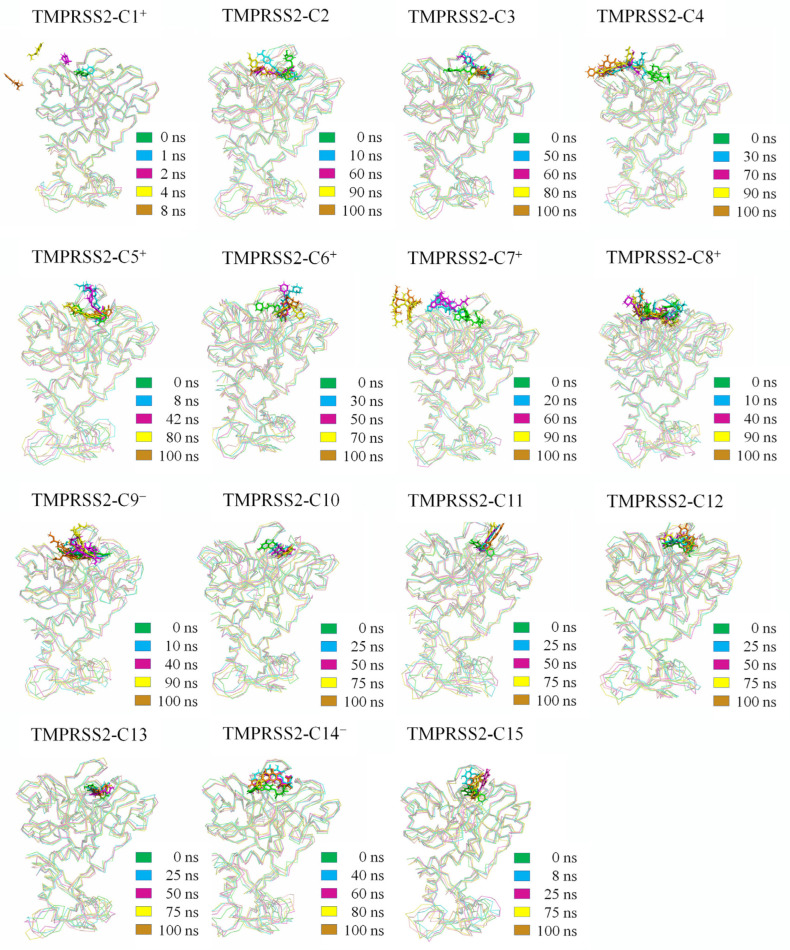
Superimposed structures of TMPRSS2–compound complexes from MD simulations. For each complex, superimposition was performed by least-square fitting of snapshots at four distinct time points to the initial structure at 0 ns. TMPRSS2 is shown in ribbon representation and compounds in stick representation, with different colors representing individual time points.

**Figure 3 ijms-26-00587-f003:**
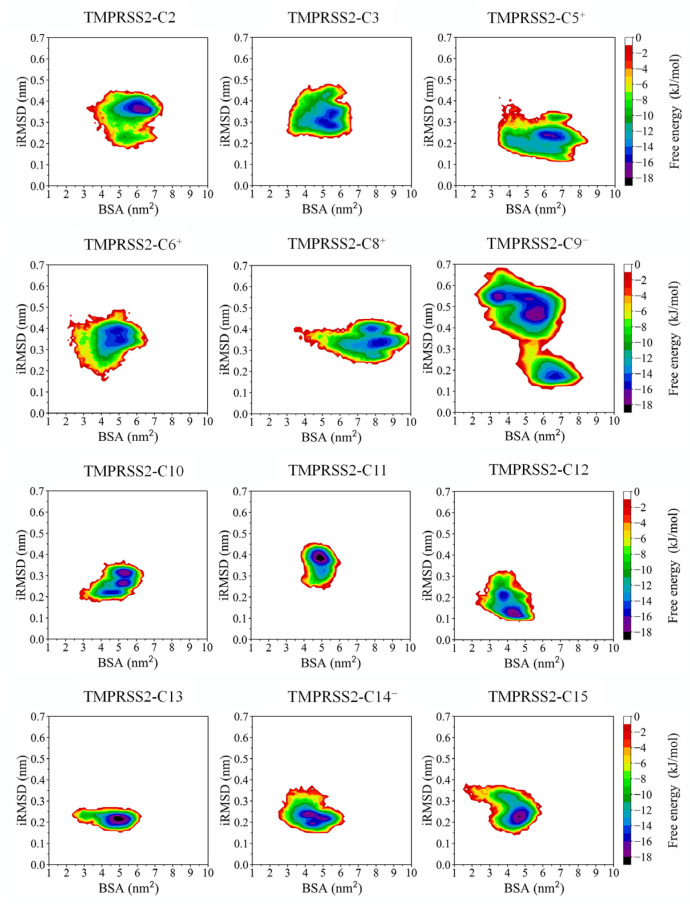
FELs for the 12 complexes constructed from equilibrated MD trajectories with BSA and iRMSD as reaction coordinates. Relative free energy (kJ/mol) is indicated by the color bar.

**Figure 4 ijms-26-00587-f004:**
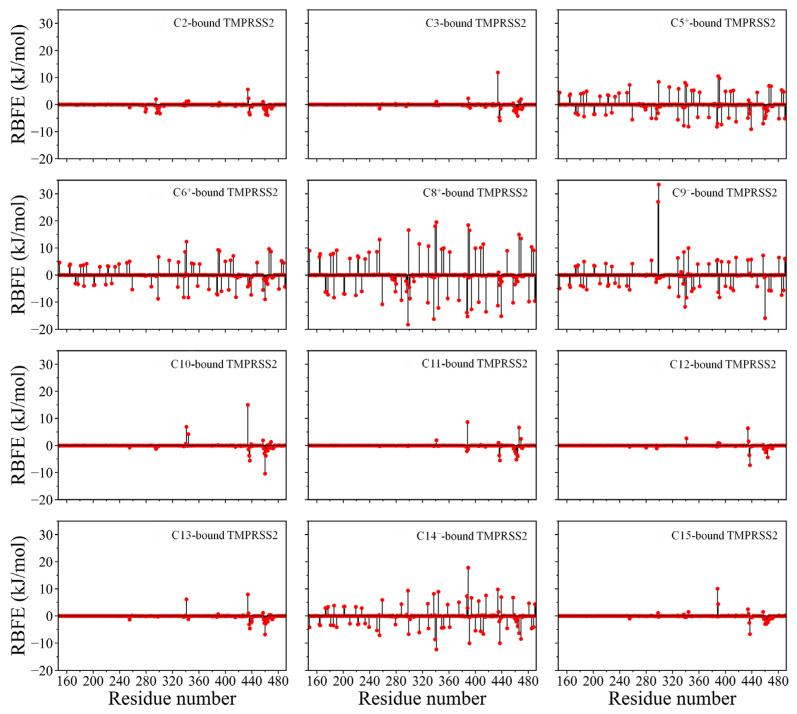
Per-RBFE values for TMPRSS2 in complex with 12 different compounds. For detailed numerical RBFE values and their energy components, refer to Appendix A.

**Figure 5 ijms-26-00587-f005:**
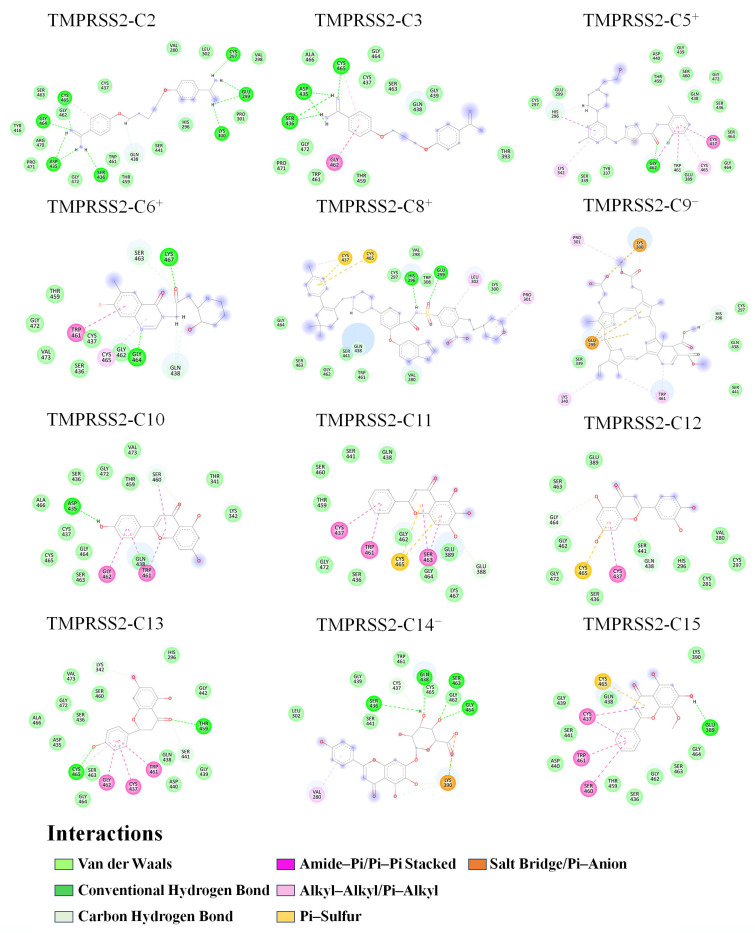
Representative 2D binding modes of the 12 complexes. TMPRSS2 residues are shown as circles, colored by the dominant interaction type. Non-vdW interactions are represented by dashed lines in different colors. The size of the light-gray circle around a residue reflects its burial extent upon binding, while the size of the blue vignette around a ligand atom indicates its degree of solvent exposure.

**Table 1 ijms-26-00587-t001:** Compound identification details and optimal docking scores with TMPRSS2.

Identifier ^a^	Name	ZINC ID	Docking Score (kcal/mol)
C1^+^	Debrisoquine ^b^	ZINC3594299	−5.13
C2	Pentamidine ^b^	ZINC1530775	−5.82
C3	Propamidine ^b^	ZINC1665564	−6.35
C4	Cilnidipine ^c^	ZINC19632657	−5.44
C5^+^	Dasatinib ^c^	ZINC3986735	−5.59
C6^+^	Halofuginone ^c^	ZINC1849658	−6.16
C7^+^	Homoharringtonine ^c^	ZINC43450324	−4.17
C8^+^	Venetoclax ^c^	ZINC150338755	−4.85
C9^−^	Verteporfin ^c^	ZINC43769730	−5.42
C10	Apigenin ^d^	ZINC3871576	−5.80
C11	Baicalein ^d^	ZINC3871633	−5.50
C12	Luteolin ^d^	ZINC18185774	−6.94
C13	Naringenin ^d^	ZINC156701	−6.27
C14^−^	Scutellarin ^d^	ZINC21992916	−5.67
C15	Wogonin ^d^	ZINC899093	−6.02

^a^ Compound identifier used in this paper; ‘+’ and ‘−’ indicate single positive and negative charges, respectively. ^b^ Identified by Peiffer et al. [42]. ^c^ Identified by Chen et al. [43]. ^d^ Identified by Huang et al. [44].

**Table 2 ijms-26-00587-t002:** Average values and SDs of the MM-PBSA energy components and BFE (kJ/mol) for the 12 complexes, calculated from their respective structural ensembles.

Complex	Δ*E*_vdw_ ^a^	Δ*E*_elec_ ^a^	Δ*G*_polar_ ^a^	Δ*G*_nonpolar_ ^a^	Δ*G*_binding_ ^b^
TMPRSS2-C2	−167.1 ± 12.1	−49.9 ± 12.8	146.7 ± 14.0	−17.3 ± 0.8	−87.6 ± 14.1
TMPRSS2-C3	−142.2 ± 11.9	−48.3 ± 6.7	144.4 ± 15.1	−14.3 ± 0.8	−60.5 ± 16.1
TMPRSS2-C5^+^	−173.9 ± 10.2	−0.7 ± 10.0	151.0 ± 20.0	−18.0 ± 1.0	−41.5 ± 18.2
TMPRSS2-C6^+^	−148.8 ± 13.3	22.0 ± 20.4	126.0 ± 33.0	−13.5 ± 1.1	−14.4 ± 21.3
TMPRSS2-C8^+^	−245.7 ± 19.3	25.2 ± 18.8	231.4 ± 28.7	−23.8 ± 1.2	−12.9 ± 29.2
TMPRSS2-C9^−^	−147.3 ± 9.7	−99.4 ± 19.3	231.7 ± 53.6	−17.5 ± 0.9	−32.5 ± 39.9
TMPRSS2-C10	−146.7 ± 10.7	−22.0 ± 5.4	114.8 ± 13.1	−14.3 ± 0.7	−68.1 ± 13.4
TMPRSS2-C11	−147.5 ± 10.4	−11.6 ± 9.8	91.1 ± 23.1	−13.1 ± 0.7	−81.2 ± 21.1
TMPRSS2-C12	−125.9 ± 9.0	−14.6 ± 8.7	96.5 ± 17.1	−12.6 ± 0.8	−56.7 ± 16.9
TMPRSS2-C13	−152.5 ± 10.8	−18.1 ± 7.2	104.3 ± 11.3	−13.2 ± 0.7	−79.4 ± 13.3
TMPRSS2-C14^−^	−125.2 ± 9.0	−114.7 ± 25.1	222.3 ± 46.7	−14.1 ± 0.8)	−31.7 ± 28.6
TMPRSS2-C15	−129.1 ± 9.9	−17.0 ± 6.2	92.5 ± 14.4	−12.2 ± 0.7	−65.8 ± 14.4

^a^ Energy components include van der Waals (Δ*E*_vdw_) and electrostatic (Δ*E*_elec_) interaction energies, as well as polar (Δ*G*_pola_) and nonpolar (Δ*G*_nonpolar_) solvation free energies. ^b^ Binding free energy (BFE) calculated as the sum of these energy components.

**Table 3 ijms-26-00587-t003:** Number of electrostatic-related intermolecular interactions in representative snapshots of the 12 TMPRSS2–compound complexes.

Complex	Salt Bridge	Hydrogen Bond		Pi-Related Stacked		Pi–Sulfur	Pi–Anion	Total ^a^
		Conventional	Carbon	Amide–Pi	Pi–Pi			
TMPRSS2-C2	0	9	1	0	0	0	0	10
TMPRSS2-C3	0	5	0	1	0	0	0	6
TMPRSS2-C5^+^	0	1	3	2	2	0	0	8
TMPRSS2-C6^+^	0	2	3	0	1	0	0	6
TMPRSS2-C8^+^	0	2	1	0	0	2	0	5
TMPRSS2-C9^−^	1	0	2	0	0	0	3	6
TMPRSS2-C10	0	1	1	2	2	0	0	6
TMPRSS2-C11	0	0	1	3	1	2	0	7
TMPRSS2-C12	0	0	2	1	0	1	0	4
TMPRSS22-C13	0	2	2	2	1	0	0	7
TMPRSS2-C14^−^	1	4	4	0	0	0	0	8
TMPRSS2-C15	0	1	0	3	1	1	0	6

^a^ Total number of electrostatic-related interactions.

## Data Availability

All data are contained within the article or its Appendix A as figures or tables.

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
