# Peer review of "Dissecting the Binding Affinity of Anti-SARS-CoV-2 Compounds to Human Transmembrane Protease, Serine 2: A Computational Study"

_ijms, 2025, doi:10.3390/ijms26020587_

Round 1
Reviewer 1 Report
Comments and Suggestions for Authors
I write this review as an experimentalist who has collaborated with several research groups doing computational studies; and we have co-published together.
I find the work of potentially high significance, but a few comments and questions regarding the methods should be further described/validated.
I could not find within the manuscript nor the supp info the PDB ID that was used as the starting coordinates for the protein structure nor what methods were utilized to equilibrate this starting condition. from my experience with my collaborators, the starting equilibration state is an important "experimtnal" aspect to get correct before analyzing binding information or trajectories that result after binding.
Can the authors please add this information so that the suitability of the starting coordinates can be verified?
Additionally, this comment follows upon my first question as to whether 100 ns is long enough time to ascertain structural fluctuations and binding. It would be good if the authors can provide some information about replicates for the computational work and references to other studies showing/verifying that the choice of 100ns is appropriate for these studies.
Especially if the protein was not pre-equilibrated, there are concerns about the overall approach, which can then significantly impact the results and conclusions. hence, I believe it is important for the authors to better clarify these initial conditions and appropriateness of the choices they have made.
Finally, I think the manuscript is well written in terms of explaining the data/results and the impact/significance of their findings. I ask that the authors provide an intro figure/scheme that shows the viral mechanisms and proteins involved. Particularly if the TMPRSS2 is a membrane anchored/associated protein, the manuscript can benefit from more description in the introduction regarding the structural work that has been done on this protein to date, as well as some discussion of the exclusion of the anchoring domain and if this has been validated experimentally as a suitable target. Oftentimes for Xray studies only the soluble domains are studied for ease of sample preparation. One would like to know if the presence/absence of the membrane domain/membrane interactions can modulate the activity of this protein. If the studies have not been done, it would still be good for the authors to comment on this potential limitation of their work.
Author Response
Please see the attached file named "response_letter_to_reviewer 1.docx"

Reviewer 2 Report
Comments and Suggestions for Authors
The article, "Dissecting TMPRSS2-Binding Affinity of Anti-SARS-CoV-2 Compounds: A Computational Study," focuses on the interaction between TMPRSS2, a critical protease facilitating SARS-CoV-2 entry, and 15 antiviral compounds. The study utilizes molecular docking, molecular dynamics (MD) simulations, free energy landscape (FEL) reconstruction, and binding free energy (BFE) calculations to evaluate the binding mechanisms and affinities of these compounds. This work offers significant contributions to understanding TMPRSS2 interactions and provides a framework for developing effective SARS-CoV-2 inhibitors.
The study employs a robust combination of docking, MD simulations, FEL analysis, and MM-PBSA to dissect protein-ligand interactions. Fuerthermore, the use of FELs to visualize binding conformations and thermodynamics adds depth to the analysis. The differentiation of vdW and electrostatic contributions offers practical guidance for designing TMPRSS2 inhibitors, and the identification of hotspot residues provides valuable targets for drug optimization.
Major Comments
While the computational findings are robust, the absence of experimental validation (e.g., in vitro assays or structural studies) limits the study's translational potential. As a practical alternative, the authors could utilize experimentally reported affinity values or kinetic constants from the literature to validate their protocol. By comparing the predicted relative binding free energy rankings with these experimental data, the authors could demonstrate the accuracy and reliability of their computational approach. This benchmarking would not only strengthen the study’s conclusions but also enhance confidence in the methodology presented.
The study appears to rely on single MD runs for each compound. Adding replicates would improve the statistical reliability of the results and reduce the risk of artifacts from initial conditions.
The study focuses on 15 compounds but does not explore the generalizability of findings to other classes of inhibitors. Discussing how these insights might extend to other compounds or chemical scaffolds would broaden the impact.
Minor Comments
Table 2 summarizing energy components need to be improved
Minor grammatical errors (e.g., inconsistent verb tense) should be addressed to improve readability.
Although the study highlights the therapeutic potential of TMPRSS2 inhibitors, a brief discussion of how the findings could inform clinical strategies or compound optimization pipelines would enhance translational value.
Author Response
Please see the file named "response_letter_to_reviewer 2.docx"

Round 2
Reviewer 1 Report
Comments and Suggestions for Authors
The authors have very nicely addressed my concerns.
I still somewhat disagree that a figure in the introduction describing the lifecycle isn't needed, as it is often times frustrating to go to other references to understand all of the "players" and proteins being discussed in an introduction.
As an experimentalist, I like to see structures of the proteins the work is talking about and what system it is involved in.
However, I can disagree with this point and still assess the paper as worthy of publication.
Reviewer 2 Report
Comments and Suggestions for Authors
I confirm that the authors have thoroughly addressed all of my suggestions and questions. I find their responses satisfactory, and the manuscript meets the standards for publication. I recommend its acceptance.